# Why Do Children in Slums Suffer from Anemia, Iron, Zinc, and Vitamin A Deficiency? Results from a Birth Cohort Study in Dhaka

**DOI:** 10.3390/nu11123025

**Published:** 2019-12-11

**Authors:** Mustafa Mahfuz, Laura E. Murray-Kolb, S. M. Tafsir Hasan, Subhasish Das, Shah Mohammad Fahim, Mohammed Ashraful Alam, Laura Caulfield, Tahmeed Ahmed

**Affiliations:** 1Nutrition and Clinical Services Division, icddr,b, Dhaka 1212, Bangladesh; tafsir.hasan@icddrb.org (S.M.T.H.); subhasish.das@icddrb.org (S.D.); mohammad.fahim@icddrb.org (S.M.F.); mashraful@icddrb.org (M.A.A.); tahmeed@icddrb.org (T.A.); 2Faculty of Medicine and Life Sciences, University of Tampere, 3310 Tampere, Finland; 3The Pennsylvania State University, University Park, PA 16802, USA; lem118@psu.edu; 4Johns Hopkins Bloomberg School of Public Health, Baltimore, MD 21205, USA; lcaulfi1@jhu.edu

**Keywords:** anemia, micronutrient deficiency, latent class growth modeling, children, Bangladesh

## Abstract

Considering the high burden of micronutrient deficiencies in Bangladeshi children, this analysis aimed to identify the factors associated with micronutrient deficiencies and association of plasma micronutrient concentration trajectories from 7 to 24 months with the concentrations at 60 months of age. Plasma samples were collected at 7, 15, 24, and 60 months of age, and hemoglobin, ferritin, zinc, and retinol concentrations of 155, 153, 154, and 155 children were measured, respectively. A generalized estimating equation was used to identify the factors associated with micronutrient deficiencies, while latent class growth modeling identified the trajectories of plasma micronutrients from 7 to 24 months and its association with the concentrations of micronutrients at 60 months was examined using multiple linear regression modeling. Early (AOR = 2.21, *p* < 0.05) and late convalescence (AOR = 1.65, *p* < 0.05) stage of an infection, low ferritin (AOR = 3.04, *p* < 0.05), and low retinol (AOR = 2.07, *p* < 0.05) were associated with increased anemia prevalence. Wasting at enrollment was associated with zinc deficiency (AOR = 1.8, *p* < 0.05) and birth weight was associated with ferritin deficiency (AOR = 0.58, *p* < 0.05). Treatment of drinking water was found protective against vitamin A deficiency (AOR = 0.57, *p* < 0.05). Higher trajectories for ferritin and retinol during 7–24 months were positively associated with plasma ferritin (β = 13.72, *p* < 0.05) and plasma retinol (β = 3.99, *p* < 0.05) at 60 months.

## 1. Introduction

Child undernutrition is the result of the interplay between multiple causal and contextual factors including poor complementary feeding which results in both macro and micronutrient deficiencies [1,2,3]. Diets of infants and young children in developing countries are usually cereal based, lacking adequately bioavailable micronutrients [4]. Micronutrients such as zinc, vitamin A, and iron are essential for growth, immunity, and cognitive development [5]. Globally, half of preschool children are anemic due mainly to iron deficiency, around 140 million preschool children have subclinical vitamin A deficiency [6], and approximately half the population is at risk of developing zinc deficiency [7].

Anemia is very common in preschool and school aged children and has implications for child nutrition, growth, and survival [8]. Vitamin A deficiency is also a public health problem, and vitamin A is essential for various physiological functions, especially tissue development, metabolism, and resistance to infections [9]. Zinc is a trace mineral that plays a vital role in cellular growth, specifically in the production of enzymes necessary for the synthesis of RNA and DNA [10]. In addition to its protective role in diarrhea and acute respiratory tract infection, zinc is also essential for child growth and cognitive development [11]. There are ample data on the micronutrient status of children under the age of five years and on the risk factors for different micronutrient deficiencies at specific ages. However, there is a paucity of data on micronutrient status from cohort studies in which repeated samples were collected from infancy to five years of age.

Factors associated with different micronutrient deficiencies include low or inadequate dietary consumption, morbidity episodes, and socio-demographic status as well as child caring practices reflected by maternal education level. Some of these determinants are fixed and some are essentially time variant. Within a child, we can track variations in micronutrient status if we collect repeated measures longitudinally over a period of several months. At the population level, there is a possibility that these changes follow specific patterns or trajectories that may predict future micronutrient status. Therefore, it is important to explore the relationship between patterns of change in micronutrient status during early childhood and micronutrient status in later years.

The Etiology, Risk Factors, and Interactions of Enteric Infections and Malnutrition and the Consequences for Child Health and Development (MAL-ED) study is a birth cohort study in which children were followed longitudinally from birth to 60 months of age [12]. At the Bangladesh site of the MAL-ED study, plasma micronutrient concentrations and hemoglobin were assessed at 7, 15, 24 and 60 months of age [13]. Most studies from low- and middle-income countries have not assessed child micronutrient status at four time points longitudinally while simultaneously collecting data on growth, nutrition, infection, and socio-demographic data.

The current analysis evaluated factors associated with the deficiencies of zinc, vitamin A, and iron, as well as anemia from seven months to five years of age and also aimed to identify whether distinct trajectories of micronutrient concentrations from 7 to 24 months predict micronutrient status at 60 months of age.

## 2. Materials and Methods

### 2.1. Study Design and Participants

This birth cohort study was conducted in a slum settlement located in Bauniabadh in the Mirpur area of Dhaka city. The area is inhabited by people with low socioeconomic status and limited sanitary conditions. The study site, location, demography, and socio-economic status have been reported previously [12,13]. In the MAL-ED birth cohort study in Bangladesh, children were enrolled, on average, 3.5 days after birth using predefined inclusion and exclusion criteria and were followed longitudinally until 60 months of age. The first child was enrolled on 11 February 2010 and the five-year follow-up was completed on 12 February 2017.

### 2.2. Inclusion and Exclusion Criteria

The MAL-ED study had well defined inclusion and exclusion criteria for enrollment. Inclusion criteria included apparently healthy infants, enrolled within 17 days after birth, parents agreed to be visited by the research staff in their household twice weekly during the study period and they did not have any plan to move outside of the study area for more than 30 days during the first 6 months of follow-up. The exclusion criteria included: family had a plan to migrate to another location; mother was less than 16 years old; the infant was not a singleton; birth weight was <1.5 Kg; child had acute or chronic clinical conditions, congenital anomalies, or developmental delays diagnosed by a physician; hospitalized after birth due to any clinical condition; and the parents failed to sign the consent form.

### 2.3. Ethical Issues

This study was approved by the Research Review Committee (RRC) and Ethical Review Committee (ERC) of icddr,b. Prior to enrollment, the parents were briefed about the study, study objectives and detailed methodology. Before any data were collected, written consent was obtained via signature from one of the parents.

### 2.4. Data Collection

After enrollment, baseline socio-demographic, feeding and morbidity data were collected and anthropometry was performed by trained research staff. The children were visited twice weekly to collect feeding and morbidity data and anthropometric measurements were taken every month until 24 months of age [14]. We used a qualitative 24 h food frequency questionnaire to collect feeding data from month 1 to month 8. From 9 to 24 months we switched to a monthly quantitative 24 h recall approach to estimate nutrient and energy intakes from non-breast milk foods. From 24 months onward, quantitative 24 h recall data were collected every six months until 60 months. This analysis used dietary intake data collected at 60 months of age using the 24 h recall method. Detailed information regarding the methodology and data collection has been published elsewhere [15]. Anthropometry data were collected on a monthly basis and data on water-sanitation and hygiene, assets, income, and food security were collected every six months [14,15].

### 2.5. Biological Sample Collection

Blood samples were collected at 7, 15, 24, and 60 months of age and plasma was obtained by centrifuging the blood. Plasma samples were stored at −80 °C until analysis [11,15].

### 2.6. Measurement of Plasma Micronutrient Status

Plasma zinc concentration was measured using flame atomic absorption spectrophotometry (Shimadzu AA-6501S, Kyoto, Japan) [16]. Plasma retinol concentration was measured by reverse phase HPLC using C18 column (Discovery C18, 25cmX4mm, 5µm, Cat# 504971) and detected at 325nm [17]. Plasma ferritin, C-reactive protein (CRP) and alpha-1-acid glycoprotein (AGP) concentrations were determined by immunoturbidimetric assay using commercial kits from Roche diagnostics on a Roche automated clinical chemistry analyzer (Hitachi −902, Boehringer Mannheim, Germany). A HemoCue 201 instrument was used to measure hemoglobin concentration [15]. Details are described in Appendix A.

### 2.7. Infection Category and Adjustment of Ferritin, Zinc and Retinol for Inflammation

Using CRP and AGP concentrations, participants were divided into four groups: “incubation” (CRP > 5 mg/L and AGP < 1 g/L), “early convalescence” (CRP > 5 mg/L and AGP > 1 g/L), “late convalescence” (CRP < 5 mg/L and AGP > 1 g/L, and the “healthy/reference” (CRP < 5 mg/L and AGP < 1 g/L) group [18]. The geometric mean of plasma ferritin, zinc, and retinol were calculated for each of the infection groups described above. Then correction factors were calculated as ratios of geometric means of the “healthy/reference” group to that of the infection groups (incubation, early convalescence, late convalescence). Ferritin, zinc, and retinol concentrations in infection groups were then adjusted by multiplying by the group specific correction factors.

### 2.8. Definitions and Cut-offs

For the purpose of statistical analyses, cut-off values for each micronutrient deficiency were: anemia (hemoglobin concentration < 11.0 g/dL), iron deficiency (plasma ferritin < 12.0 ng/mL), zinc deficiency (plasma zinc < 9.9 mmol/L), and vitamin A deficiency (plasma retinol < 20 μg/dL) [19,20,21,22].

The Water-sanitation-hygiene, Asset status, Maternal education status, and monthly Income (WAMI) index is a composite score for assessing socioeconomic status. This index is widely used in MAL-ED publications and the methodology is published elsewhere [23]. To examine the factors associated with the different micronutrient deficiencies we have not used the WAMI index, rather we used each of the components of WAMI as exposure variables. However, the WAMI index at 60 months was used as one of the explanatory variables to explore the association of trajectories of micronutrient concentrations from 7 to 24 months with micronutrient status at 60 months of age.

Improved toilet was defined per WHO guidelines: presence of flush latrine connected to sewer system, septic tank, pit latrine; ventilated improved pit latrine; pit latrine with slab or composting toilet [24].

Low birth weight: A birth weight of <2.5 kg was considered as low birth weight; those weighing <1.5 kg were excluded from the study. Although study children in the MAL-ED Bangladesh site were enrolled, on average, 3.5 days after birth, their original birth weight was recorded from the delivery/birth certificates.

Household food security access status was categorized by using the Household Food Insecurity Access Scale (HFIAS) developed by the Food and Nutrition Technical Assistance (FANTA) project [25]. Based on individual household food security access scores, the food security status was categorized to households with no food insecurity, mild food insecurity, moderate food insecurity, and severe food insecurity. We have used enrollment HFIAS for this analysis.

Asset index: The household asset index was constructed using household asset data obtained from the SES questionnaire. From asset related dichotomous variables, using principal components analysis in STATA software, a common factor score for each household was produced. After ranking by their score, we divided first principal component score into quintiles to create five categories where the first category represents the poorest and the fifth category represents the wealthiest ones.

### 2.9. Statistical Analysis

We examined the distribution of micronutrient concentrations by using histograms, box plots, Q-Q plots, or frequency tables as appropriate. Characteristics of the study participants at baseline were reported as mean and standard deviation for continuous variables and frequency distributions for categorical variables. Those that were not normally distributed were described by median and inter-quartile range (IQR). Continuous variables with normal distributions were compared between groups using Student’s t-test after verifying the equality of variance (Levene’s test). The difference in proportion was compared using a Chi-square test or the Fisher’s exact test if the expected number in any cell was <5.

To investigate factors associated with anemia, iron deficiency, zinc deficiency and vitamin A deficiency from birth to five years of age, we used generalized estimating equation (GEE) regression. Our outcome variables were binary categorical variables measured at 7, 15, 24, and 60 months. GEE is a population specific method based on average changes in response over time and the impact of covariates on these changes. Therefore, the reported odds ratio is a pooled odds ratio of the effect of all the predictors over 7 to 60 months of age. We assumed an autoregressive (AR) covariance matrix with robust variance estimates. Initially, bivariate analysis was performed to identify the unadjusted effect of each predictor on each deficiency through individual GEE models. The variable selection was based on availability of data at different time points (Appendix A). Multi-collinearity between independent variables was examined using correlation matrix and variance inflation factor (VIF) values. In succeeding models, covariates whose p-values were less than 0.20 in bivariate analysis with the outcome variable were entered simultaneously to obtain the adjusted final model. The best models were selected based on the lowest quasi-likelihood under independence model criterion (QIC) value. A probability of less than 0.05 was considered statistically significant and the strength of association was determined by estimating the adjusted odds ratios (AOR) and their 95% confidence intervals (CIs).

We used latent class growth modeling (LCGM) to identify distinct clusters of children following similar trajectories with regard to the pattern of hemoglobin, ferritin, retinol, and zinc at 7, 15, and 24 months [26,27]. We built separate trajectory models for hemoglobin, ferritin, retinol, and zinc. The analyses were restricted to the children for whom data on the outcomes were available at 7, 15, and 24 months (three time-points) [28]. In addition, for ferritin, three very high (unusual) values were dropped from the dataset. The sample sizes for LCGM were reduced to 155 for hemoglobin, 153 for ferritin, 154 for retinol, and 155 for zinc. We selected the final models with optimal number and shape of trajectories based on Bayesian information criteria (BIC), log Bayes factor, the statistical significance of quadratic terms, whether 95% confidence intervals of trajectories overlapped, and the percentage of the population in each trajectory group [29]. After selecting the final model, we calculated the posterior probabilities for each individual of belonging to each of the trajectory groups, and individuals were assigned to a trajectory group based on the maximum-probability assignment rule [30]. We reported the findings of the LCGM following the GRoLTS-Checklist: Guidelines for Reporting on Latent Trajectory Studies [31]. A detailed description of LCGM methodology and STROBE checklist for reporting cohort studies are included in Appendix A.

In subsequent analyses, multiple linear regression models with robust standard errors were fitted to examine the association of concentrations of hemoglobin, plasma ferritin, retinol, and zinc at the age of 60 months with the trajectories identified through LCGM. As zinc was found to have a single trajectory (described in detail in the results section), zinc concentrations at 24 months were used instead of any trajectory to assess the predictive association with level of zinc at 60 months. We considered several covariates collected at the age of 60 months in building these models which include mean intakes of energy intake (kcal/d), protein (g/d), fat (g/d), carbohydrates (g/d), iron (mg/d), vitamin A (ug_RE/d), and zinc (mg/d). We also considered the phytate to iron ratio and the phytate to zinc ratio of the diet, as well as the percent of energy from carbohydrates (minus fiber), protein, and fat. Non-dietary covariates included socioeconomic and WASH variables, the WAMI score, and child sex. Details on the methodology are described in Appendix A. The final regression models included only iron intake, the percentage of energy from protein, and WAMI score for hemoglobin, level of zinc at 24 months, total energy intake, protein intake, and ferritin, vitamin A, retinol, and zinc intake. Due to missing values for the outcome variables, the sample sizes for the linear models were reduced to 142 for hemoglobin, 138 for ferritin and retinol, and 140 for zinc.

The statistical analyses related to LCGM were performed using the “traj plugin” in Stata (StataCorp, College Station, Texas 77845 USA, version 14.1) [32], a Stata equivalent of the widely used “proc traj” in SAS [28]. Outputs of LCGM models were plotted using “traj” in Stata and the R packages “lcmm” and “ggplot2” in R (version 3.5.1). All other statistical analyses were performed with Stata/PC (StataCorp, College Station, Texas 77845 USA, version 14.1).

## 3. Results

### 3.1. Basic Socio-Demographic Characteristics

At enrollment, data were available for 212 newborns. Male children comprised 47% of the group and 22% had low birth weight (<2.5 kg). The mean age of the mothers was 24.8 years and 45% of them were either illiterate or did not complete the primary level of education. Three fourths of the participants were from food secure households and their average monthly family income was 120 USD. Sixty percent of the families used treated drinking water by any means and two-thirds of them had improved toilet facilities in their households. Prevalence of stunting was 17% at enrollment, and 30% had inflammation at 7 months, defined by high concentrations of acute phase proteins (Table 1).

### 3.2. Plasma Micronutrient Status

Results of the assays done for hemoglobin, plasma ferritin, plasma zinc, and plasma retinol are reported in Figure 1.

Using the predefined cut-off values, it was observed that anemia, zinc deficiency, and vitamin A deficiency declined gradually from month 7 to month 60 whereas iron deficiency (low plasma ferritin) increased gradually from 7 months to 24 months and was then markedly reduced at 60 months of age. All of these changes were found to be statistically significant (*p* < 0.05) (Table 2).

#### 3.2.1. Factors Associated with Anemia from 7 to 60 Months

Bivariate analyses showed that infection status, zinc deficiency, iron deficiency and vitamin A deficiency were positively associated with anemia. On the other hand, age, birth weight, monthly income and higher education of the mother were negatively associated with anemia (*p* < 0.05) (Appendix A). Multivariable analysis revealed that early (AOR: 2.21, 95% CI 1.14–4.27, *p* < 0.05) or late (AOR: 1.65, 95% CI 1.03–2.64, *p* < 0.05) convalescence stage of an infection, iron deficiency (AOR: 3.04, 95% CI 2.08–4.44, *p* < 0.05), and vitamin A deficiency (AOR: 2.07, 95% CI 1.31–3.28, *p* < 0.05) were associated with an increased prevalence of anemia. Whereas, age (AOR: 0.94, 95% CI 0.92–0.96, *p* < 0.05), female sex (AOR: 0.49, 95% CI 0.32–0.77, *p* < 0.05), higher maternal education (AOR: 0.38, 95% CI 0.15–0.92, *p* < 0.05) compared to no education were negatively associated with anemia (Table 3).

#### 3.2.2. Factors Associated with Iron Deficiency (Low Ferritin) from 7 to 60 Months

In bivariate analyses, age of the child, maternal age, birth weight, and female sex were associated with lower odds of iron deficiency, and being underweight at enrollment (weight-for-age z-score < −2), and anemia and vitamin A deficiency were associated with increased odds of iron deficiency (Appendix A). On the other hand, multivariable analysis showed that age in months (AOR: 0.98, 95% CI 0.97–0.98, *p* < 0.05), female sex (AOR 0.65, 95% CI 0.47–0.97, *p* < 0.05), and birth weight (AOR 0.67, 95% CI 0.38–0.88, *p* < 0.05) were associated with lower risk of iron deficiency and anemia (AOR 2.75, 95% CI 1.87–4.05, *p* < 0.05), and higher maternal education (AOR: 2.40, 95% CI 1.08–5.34, *p* < 0.05) was associated with greater risk iron deficiency (Table 3).

#### 3.3.3. Factors Associated with Zinc Deficiency from 7 to 60 Months

Bivariate analyses revealed that zinc deficiency was positively associated with anemia and the condition improved with an increase in age in months (Appendix A). After controlling for other explanatory variables, multivariable analysis showed that age in months (AOR: 0.95, 95% CI 0.93–0.98, *p* < 0.05) was negatively associated with zinc deficiency, and wasting (weight-for-length z-score <−2) at enrollment (AOR: 1.86, 95% CI 1.11–3.11, *p* < 0.05) was positively associated with risk of zinc deficiency (Table 3).

#### 3.3.4. Factors Associated with Vitamin A Deficiency from 7 to 60 Months

Anemia, low plasma zinc and severe household food insecurity (compared to children from food secure households) were associated with increased odds of vitamin A deficiency, and age in months, monthly income, and higher socio-economic status were protective of vitamin A deficiency in bivariate analyses (Appendix A). In the multivariable model, age (AOR: 0.96, 95% CI 0.95–0.98, *p* < 0.05) and treatment of drinking water (AOR: 0.57, 95% CI 0.36–0.89, *p* < 0.05) were significantly associated with lower odds of vitamin A deficiency, and anemia (AOR: 1.69, 95% CI 1.06–2.69, *p* < 0.05) was associated with increased odds of vitamin A deficiency (Table 3).

### 3.3. Latent Class Growth Model to Identify Group Trajectories of Plasma Micronutrients

Based on the criteria described in the Statistical Analysis Section (BIC, log Bayes factor approximation), the difference in the population distribution of the progression of hemoglobin concentration was best characterized by a two-class model with quadratic components for both trajectories. Group 1 indicated an overall decreasing pattern over the months with mean hemoglobin concentrations of 10.31 g/dL, 9.78 g/dL, and 10.16 g/dL at 7, 15, and 24 months, respectively, while Group 2 indicated an overall increasing pattern with mean hemoglobin concentrations of 11.36 g/dL, 11.83 g/dL, and 12.36 g/dL at 7, 15, and 24 months, respectively (Table 4 and Figure 2).

For ferritin, a two-class model with quadratic components for both trajectories was retained as the final and most parsimonious model. Mean ferritin concentration for the children belonging to Group 1 showed an overall lower value over the months, whereas Group 2 had higher mean values. Mean ferritin concentration in Group 1 was 24.2 ng/mL, 13.19 ng/mL, and 13.99 ng/mL, and in Group 2 was 82.97 ng/mL, 21.18 ng/mL, and 18.21 ng/mL at 7, 15, and 24 months, respectively (Table 4 and Figure 3).

Likewise, for retinol, a two-class model with quadratic components for both trajectories was retained as the final and most parsimonious model. Mean retinol concentration for the children belonging to Group 1 showed overall lower mean values over the months whereas Group 2 had higher mean values. Mean retinol concentration in Group 1 was 21.47 µg/dL, 25.27 µg/dL, and 24.33 µg/dL, and in Group 2 was 26.8 µg/dL, 44.01 µg/dL, and 30.45 µg/dL at 7, 15, and 24 months, respectively (Table 4 and Figure 4).

In contrast, a one-trajectory model with a quadratic component was retained as the final model for zinc. The more complex models with two, three, or four trajectories generated trajectories with insufficient cluster size (less than 5% of the study population). This result indicates that no significant distinct classes existed in the population for the progression of concentration of plasma zinc and all individuals followed a similar pattern over the months. Mean zinc concentration among the children included in the analysis was 11.71 mmol/L, 11.53 mmol/L, and 12.3 mmol/L at 7, 15, and 24 months, respectively (Table 4 and Figure 5).

For all the subgroups of hemoglobin, retinol, and zinc, the average posterior probability of group membership was above 0.8, odds of correct classification was more than 5, and modeled group probabilities were in good agreement with the proportions of group assignment following the maximum-probability assignment rule (Appendix A).

### 3.4. Multiple Linear Regression Models to Examine Association of Group Trajectories with Micronutrient Concentrations at 60 Months

In multiple linear regression models, we observed that mean iron intake (coefficient: 0.29, 95% CI 0.04–0.55) and percent energy from protein (coefficient: 0.45, 95% CI 0.18–0.73) at 60 months were significantly associated with hemoglobin concentration at 60 months (*p* < 0.05). However, we did not find any significant association of hemoglobin at 60 months with the identified trajectories that the children’s hemoglobin concentrations followed during 7–24 months. The children whose plasma ferritin concentrations followed a higher trajectory during 7–24 months were found to have higher plasma ferritin concentration at 60 months (coefficient 13.72, 95% CI 1.15–26.28, *p* < 0.05). Plasma zinc concentration at 24 months was associated with plasma ferritin concentration at 60 months (coefficient 1.98, 95% CI 0.24–3.71, *p* < 0.05). We also observed that the children whose plasma retinol concentrations followed a higher trajectory during 7–24 months had significantly higher plasma retinol level at 60 months (coefficient 3.99, 95% CI 1.04–6.94, *p* < 0.05). Mean vitamin A intake from food at 60 months was associated with plasma retinol concentration at 60 months (*p* < 0.05). One mmol/L increase in plasma zinc at 24 months was associated with 0.21 mmol/L greater plasma zinc concentration at 60 months (Table 5). Summary information generated from this study are recorded in Table 6.

## 4. Discussion

This analysis showed a high burden of anemia, and deficiencies of iron, zinc, and vitamin A during early childhood in the Mirpur area of Dhaka followed by a markedly lower prevalence of all deficiencies at 60 months. Due to the unavailability of reliable, published longitudinal data on micronutrient status, we were unable to make any comparison with the changes we have observed in our study. According to the Bangladesh National micronutrient survey 2011, among children aged 6–60 months, 33% were anemic, 3.9% had iron deficiency, 20.5% had vitamin A deficiency, and 44.6% had zinc deficiency at the national level [33,34]. A recent review of the micronutrient status of under-five year South Asian children also reported similar burdens of anemia and iron, zinc, and vitamin A deficiencies [35].

Regarding the factors associated with anemia, and deficiencies of iron, zinc, and Vitamin A from the children followed longitudinally from 7 months to 60 months, we observed that infection, low ferritin, and low retinol were associated with a higher prevalence of anemia in multivariable analysis. In contrast, age, monthly income, and higher education of mothers compared to no education were associated with a lower risk of anemia. Iron is required to enhance immunity to prevent and protect the host from infections [36]. A previous MAL-ED multi-country pooled analysis also showed that the detection of pathogens and illness were inversely related to hemoglobin concentration [37]. Epidemiological studies also observed a positive association between vitamin A deficiency and anemia [38]. It can be explained through experimental studies that vitamin A deficiency is associated with increased hepcidin expression which directly acts on hepatic mobilization of iron stores essential for erythropoiesis [39]. As for iron deficiency, age in months, female sex, and birth weight were associated with higher plasma ferritin and anemia, and higher maternal education was associated with lower ferritin concentrations. We do not know the reason for a positive association between iron deficiency and higher maternal education. Perhaps this was due to the fact that a small number of mothers had secondary or higher level of education (4.7% of 212 mothers of children at 7 months). However, this research question should be followed up in a study consisting of both slum and non-slum children.

We also observed that age in months was negatively associated with zinc deficiency while wasting at birth was positively associated with zinc deficiency. Similarly, age and treatment of drinking water were significantly associated with lower risk of vitamin A deficiency, and anemia was associated with great risk of deficiency.

Considering the interrelation between different micronutrients, from 7 months to 60 months of age, low ferritin and low retinol concentrations were associated with greater odds of anemia. This was also evident in the Bangladesh national micronutrient survey of 2011, where higher serum ferritin was found to be associated with higher levels of hemoglobin in children less than five years of age [34]. They also reported a positive association between serum retinol and hemoglobin concentrations where the prevalence of anemia was 33% higher in children with low retinol concentrations [34]. Several other studies also found similar results [40,41]. Vitamin A most likely reduced anemia by lowering infection, augmenting erythropoiesis and releasing stored iron from the liver [34]. Our analysis also suggested a positive association between plasma zinc status at 24 months and plasma ferritin status at 60 months. If zinc is increasing erythropoiesis, then more iron will be taken out of ferritin to “keep up” with the erythropoietic needs and, therefore, the relation between plasma zinc and ferritin might be negative [42,43]. The positive relation that we see is likely due to intake where foods that are high in zinc are also high in iron. Despite the fact that zinc and iron compete for a shared absorptive pathway, iron generally does not hamper zinc absorption except at very high iron to zinc ratios [44].

We identified distinct group trajectories of hemoglobin and several plasma micronutrients between 7 months to 24 months of age and examined their predictive role in the micronutrient status at five years of age. Hemoglobin trajectory in early life did not predict the presence or absence of anemia in later childhood. This is perhaps because hemoglobin is a composite of several constituents that can be affected in different ways that can make it unpredictable. We observed that concurrent consumption of iron-rich food and protein are associated positively with hemoglobin at 60 months. Other studies also observed a positive association of consuming iron rich food and animal protein with hemoglobin status [45]. In contrast, higher concentrations of ferritin, retinol, and zinc at 24 months were each associated with concentrations at 60 months. Our findings suggest that children whose plasma ferritin concentration was around 20 ng/mL at ages 15 and 24 months continued to have better plasma ferritin level at 60 months. In our study, children who had a plasma retinol concentration of around 30 µg/dL at ages 7 and 24 months went on to have higher plasma retinol level at 60 months. This may indicate that ensuring adequate micronutrient status in the first two years of life is important for micronutrient status at five years of age, or that dietary intakes track to a certain degree over time. The consumption of micronutrients was very poor in this population. Recently, we conducted a study in the same community to explore the overall quality of the diet and calculated the nutrient adequacy ratios (NARs) for different micronutrients including vitamin A, iron, and zinc from 9 months to 24 months of age. The NAR was calculated as the ratio of average daily intake and the recommended dietary allowance of that specific nutrient. The NAR values for vitamin A, iron, and zinc were 0.21, 0.23, and 0.42, respectively, well below a ratio of 1 for adequate intake [4].

Considering the water, sanitation and hygiene (WASH) practices, we observed that children from families who treat drinking water by any means had lower risk of vitamin A deficiency. Treatment of drinking water has a direct role in preventing diarrhea and subclinical infections which can lower serum retinol concentrations by as much as 25% [46]. We did not find an association of other WASH practices with any of the outcomes. No association was observed between the overall socio-economic status and micronutrient status among the children as WAMI-score was statistically insignificant in all the analyses.

Limitations of this analysis include a reduced number of plasma micronutrient data and the unavailability of data on morbidity, enteropathy biomarkers, and pathogen burden at 60 months.

## 5. Conclusions

The results point to the importance of the first 1000 days of life. A rich endowment of micronutrients in early life is important to sustain the requirements of pre-school age. It is, therefore, crucial to have a well-nourished infant at birth through improvement of maternal health, to start complementary feeds at the optimum time and to have a diet with sufficient nutrient density to ensure adequacy.

## Figures and Tables

**Figure 1 nutrients-11-03025-f001:**
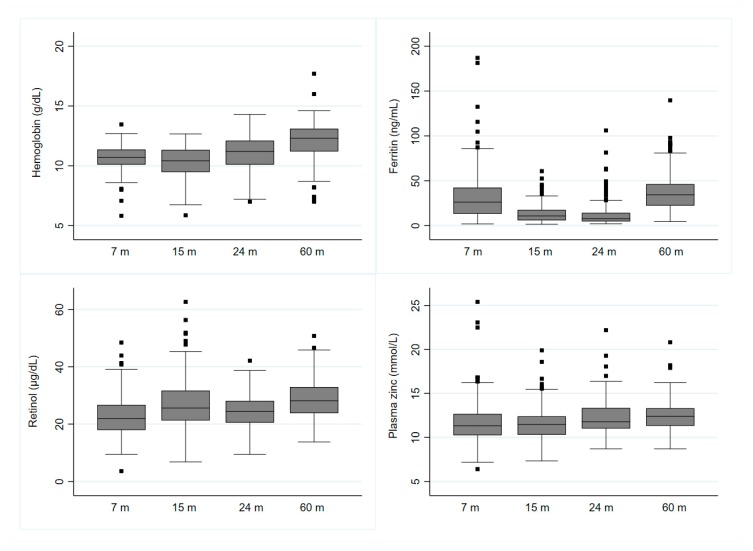
Hemoglobin and plasma micronutrient concentrations from 7 to 60 months.

**Figure 2 nutrients-11-03025-f002:**
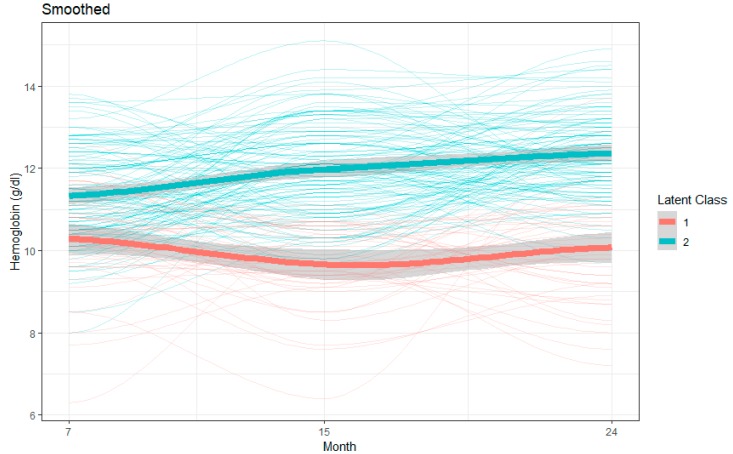
Latent growth class modeling (LCGM) derived latent trajectories with 95% confidence interval with individual trajectories of hemoglobin during age 7 to 24 months.

**Figure 3 nutrients-11-03025-f003:**
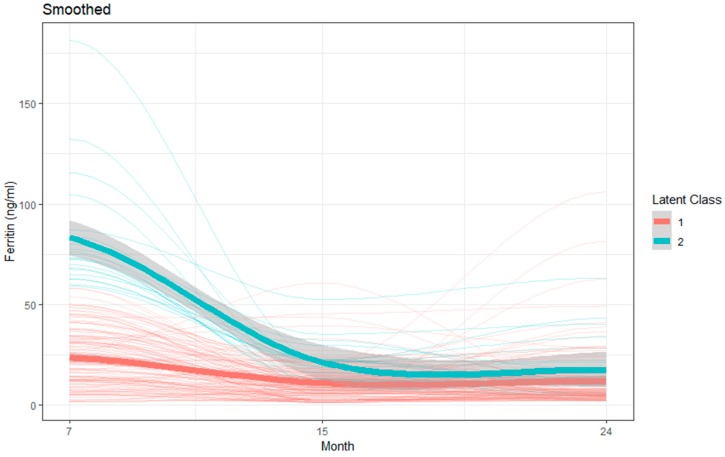
LCGM derived latent trajectories with 95% confidence interval with individual trajectories of ferritin during age 7 to 24 months.

**Figure 4 nutrients-11-03025-f004:**
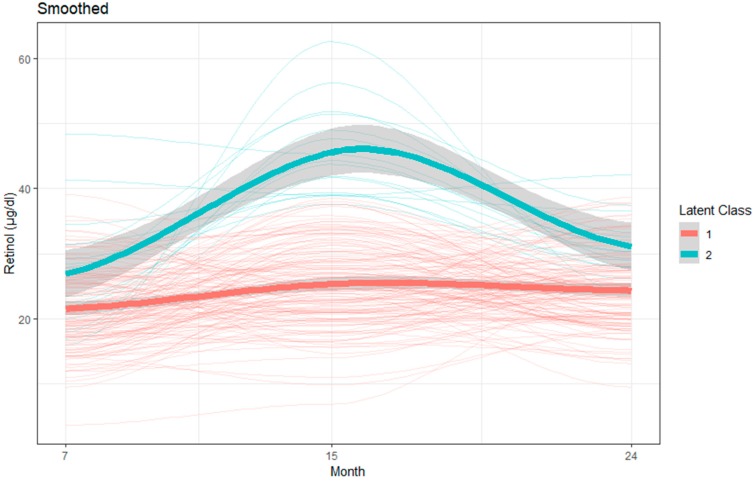
LCGM derived latent trajectories with 95% confidence interval with individual trajectories of retinol during age 7 to 24 months.

**Figure 5 nutrients-11-03025-f005:**
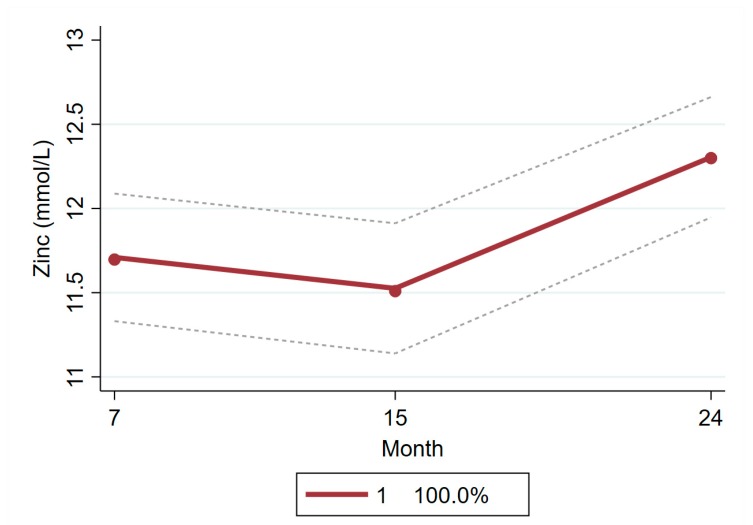
LCGM derived latent trajectory with 95% confidence interval of zinc during age 7 to 24 months.

**Table 1 nutrients-11-03025-t001:** Baseline characteristics of the study population.

Characteristics	N = 212 (%/mean ± sd)
Male	47.6
Birth weight, kg	2.8 ± 0.4
Low birth weight (<2.5 kg)	22.17
Total days of exclusive breastfeeding	106.5 ± 57.4
Maternal age	24.8 ± 4.9
Maternal education	
None	19.3
Some primary	25.9
Primary complete	18.4
Some secondary	31.6
Secondary complete or higher	4.7
Treated drinking water	60.3
Improved toilet ^a^	76.0
Hand wash after helping the child defecate	72.7
Hand wash before preparing food	22.3
Hand wash after using toilet	75.6
Food insecurity ^b^	
Secure	74.5
Mild insecurity	6.13
Moderate insecurity	11.8
Severe insecurity	7.6
Monthly income (US$), median (IQR)	97.6 (73.1, 121.9)
Asset Index ^c^	
Poorest	20.9
Poor	19.4
Middle	20.4
Wealthier	20.4
Wealthiest	18.9
Stunted (length-for-age z-score <−2)	16.9
Underweight (weight-for-age z-score <−2)	21.2
Wasted (weight-for-length z-score <−2)	16.9

^a^ Improved toilet was defined as per WHO guidelines: presence of flush latrine to piped sewer system, septic tank; pit latrine; ventilated improved pit (VIP) latrine; pit latrine with slab; or composting toilet [23]. ^b^ Household food security access status was categorized by using the Household Food Insecurity Access Scale (HFIAS) developed by the Food and Nutrition Technical Assistance (FANTA) project [24]. ^c^ Asset index: The household asset index was constructed using household asset data obtained from the SES questionnaire. From these asset related dichotomous variables, we used polychoric principal components analysis in STATA software to produce a common factor score for each household. After ranking by their score, we divided the first principal component score into quintiles to create five categories of which the first category represents the poorest and fifth category represents the wealthiest.

**Table 2 nutrients-11-03025-t002:** Anemia and adjusted * iron, zinc, and vitamin A deficiency at different time points.

	Cut-Off	Percent (n)			
		7 Months	15 Months	24 Months	60 Months
Anemia	Hb < 11 g/dL	64.3 (117)	63.9 (106)	41.7 (68)	18.8 (31)
Low Ferritin	<12 ng/mL	20.3 (42)	54.1 (105)	68 (119)	4.9 (9)
Low Zinc	<9.9 mmol/L	21.5 (44)	14.9 (29)	1.1 (2)	3.8 (7)
Low Retinol	<20 µg/dL	39.0 (80)	19.2 (37)	21.7 (38)	6.7 (12)

* Adjusted for inflammation.

**Table 3 nutrients-11-03025-t003:** Multivariable associations with anemia, low plasma ferritin, zinc, and retinol from 7 months to 60 months.

	Anemia		Zinc Deficiency		Vitamin A Deficiency		Iron Deficiency	
	AOR (95% CI)	*p*-Value	AOR (95% CI)	*p*-Value	AOR (95% CI)	*p*-Value	AOR (95% CI)	*p*-Value
Inflammation group								
Non-inflammed	Reference							
Incubation	2.43 (0.74, 7.94)	0.142						
Early convalescence	2.21 (1.14, 4.27)	0.019						
Late convalescence	1.65 (1.03, 2.64)	0.037						
Age in months	0.94 (0.92, 0.96)	<0.0001	0.95 (0.93, 0.98)	0.001	0.96 (0.95, 0.98)	<0.0001	0.98 (0.97, 0.98)	<0.0001
Female	0.49 (0.32, 0.77)	0.002					0.65 (0.47, 0.97)	0.013
Birth weight	0.67 (0.39, 1.14)	0.136			0.61 (0.36, 1.04)	0.072	0.58 (0.38, 0.88)	0.010
Wasting at birth			1.86 (1.11, 3.11)	0.018				
Food insecurity access								
Food secure					Reference			
Mild food insecurity					1.08 (0.42, 2.80)	0.87		
Moderate food insecurity					0.58 (0.29, 1.12)	0.105		
Severe food insecurity					1.49 (0.69, 3.19)	0.302		
Monthly income	0.99 (0.99, 1.00)	0.545	1.00 (0.99, 1.00)	0.079	0.99 (0.99, 1.00)	0.168		
Maternal education								
No	Reference		Reference				Reference	
Some primary	1.14 (0.63, 2.09)	0.661	0.69 (0.35, 1.37)	0.29			1.11 (0.71, 1.73)	0.65
Primary complete	1.17 (0.59, 2.30)	0.660	0.78 (0.36, 1.69)	0.53			1.29 (0.77, 2.16)	0.33
Some secondary	0.71 (0.39, 1.28)	0.250	0.53 (0.27, 1.0)	0.072			0.95 (0.62, 1.46)	0.814
Secondary complete or higher	0.38 (0.15, 0.92)	0.032	0.28 (0.06, 1.25)	0.096			2.40 (1.08, 5.34)	0.031
Treated drinking water					0.57 (0.36, 0.89)	0.016		
Improved toilet			0.67 (0.39, 1.16)	0.15	1.63 (0.98, 2.70)	0.061		
Hand wash after helping the child defecate	1.63 (0.93, 2.86)	0.089	1.33 (0.65, 2.75)	0.44				
Hand washing before food preparation	1.11 (0.66, 1.85)	0.700					1.47 (0.99, 2.17)	0.052
Hand washing after using toilet	1.44 (0.82, 2.52)	0.200	1.54 (0.79, 3.00)	0.21	0.79 (0.46, 1.38)	0.421		
Anemia			0.81 (0.49, 1.31)	0.39	1.69 (1.06, 2.69)	0.025	2.75 (1.87, 4.05)	<0.0001
Low zinc	0.79 (0.46, 1.38)	0.420			1.28 (0.71, 2.32)	0.406		
Low ferritin	3.04 (2.08, 4.44)	0.0001			1.16 (0.77, 1.75)	0.476		
Low retinol	2.07 (1.31, 3.28)	0.002	1.36 (0.78, 2.38)	0.27			0.81 (0.56, 1.19)	0.288

Explanatory variables with significance level of *p* < 0.2 in bivariate association with dependent variable were regressed in the multivariable model using generalized estimating equation (GEE). AOR: Adjusted odds ratio.

**Table 4 nutrients-11-03025-t004:** Trajectory-wise estimated mean with 95% confidence interval for hemoglobin, ferritin, retinol and zinc at different ages derived using latent class growth modeling (LCGM).

Trajectory Group	Age = 7 Months	Age = 15 Months	Age = 24 Months
Hemoglobin (g/dL)			
Group 1 (Decreasing)	Mean = 10.31; 95% CI = 9.87, 10.75	Mean = 9.78; 95% CI = 9.29, 10.27	Mean = 10.16; 95% CI = 9.56, 10.77
Group 2 (Increasing)	Mean = 11.36; 95% CI = 11.12, 11.60	Mean = 11.83; 95% CI = 11.64, 12.02	Mean = 12.36; 95% CI = 12.07, 12.65
Ferritin (ng/mL)			
Group 1 (Lower)	Mean = 24.20; 95% CI = 20.76, 27.65	Mean = 13.19; 95% CI = 10.31, 16.07	Mean = 13.99; 95% CI = 10.99, 16.99
Group 2 (Higher)	Mean = 82.97; 95% CI = 75.01, 90.93	Mean = 21.18; 95% CI = 12.85, 29.51	Mean = 18.21; 95% CI = 9.64, 26.78
Retinol (µg/dL)			
Group 1 (Lower)	Mean = 21.47; 95% CI = 20.32, 22.62	Mean = 25.27; 95% CI = 24.0, 26.55	Mean = 24.33; 95% CI = 23.25, 25.41
Group 2 (Higher)	Mean = 26.80; 95% CI = 23.58, 30.02	Mean = 44.01; 95% CI = 38.54, 49.48	Mean = 30.45; 95% CI = 26.98, 33.93
Zinc (mmol/L)			
All children	Mean = 11.71; 95% CI = 11.33, 12.09	Mean = 11.53; 95% CI = 11.14, 11.91	Mean = 12.30; 95% CI = 11.95, 12.66

**Table 5 nutrients-11-03025-t005:** Association of sixty months plasma micronutrient concentration and group trajectories of the same micronutrients from 7 to 24 months.

Outcome	Explanatory Variables	Coefficient, 95% CI	*p* Value
Hemoglobin level at 60 months	Trajectories for Hb, 7 to 24 months		
	Group 1 (Decreasing)	Reference	
	Group 2 (Increasing)	0.21 (−0.56, 0.97)	0.54
	Mean iron intake at 60 mo in mg	0.29 (0.04, 0.55)	0.022
	Mean percent energy from protein at 60 mo	0.45 (0.18, 0.73)	0.002
	WAMI	−2.63 (−5.52, 0.25)	0.073
Adjusted plasma ferritin at 60 months	Trajectories for ferritin, 7 to 24 months		
	Group 1 (Lower)	Reference	
	Group 2 (Higher)	13.72 (1.15, 26.28)	0.033
	Plasma zinc (mmol) at 24 months	1.98 (0.24, 3.71)	0.026
	Mean energy intake at 60 mo (kcal)	0.002 (−0.02, 0.02)	0.84
	Mean protein intake at 60 mo	0.09 (−0.48, 0.65)	0.76
	WAMI	−4.65 (−27.12, 17.82)	0.68
Adjusted plasma retinol at 60 months	Trajectories for retinol, 7 to 24 months		
	Group 1 (Lower)	Reference	
	Group 2 (Higher)	3.99 (1.04, 6.94)	0.008
	Mean vitamin A intake at 60 mos in µgRE	0.003 (0.00007, 0.005)	0.004
	WAMI	4.48 (−3.75, 12.73)	0.28
Adjusted plasma zinc at 60 months	Zinc (mmol/L), 24 months		
	All children	0.21 (0.02, 0.39)	0.035
	Mean zinc intake at 60 mo in mg	0.02 (−0.18, 0.21)	0.87
	WAMI	−0.17 (−1.60, 1.27)	0.82

**Table 6 nutrients-11-03025-t006:** Key information generated on plasma micronutrient status.

Factors Associated with Micronutrient Deficiencies from 7 Months to 60 Months	
Anemia	Early and late convalescence of an acute infection, and low plasma ferritin and retinol concentrations were associated with higher odds of anemia. Age, female sex, and higher maternal education were associated with reduced anemia.
Iron deficiency	Anemia was associated with higher iron deficiency. Age in months, female sex and birth weight were associated with reduced iron deficiency.
Zinc deficiency	Wasting at birth was associated with higher odds of zinc deficiency. Age in months was associated with reduced zinc deficiency.
Vitamin A deficiency	Anemia was associated with an increased vitamin A deficiency. Age in months and treatment of drinking water were associated with reduced odds of vitamin A deficiency.
Early life trajectories of micronutrient that can predict 60 months micronutrient concentrations	
Plasma ferritin concentration at 60 months	The higher trajectory for plasma ferritin during 7 to 24 months was associated with higher plasma ferritin at 60 months.
Plasma zinc concentration at 60 months	One mmol/L increase in plasma zinc concentration at 24 months was associated with 0.21 mmol/L greater plasma zinc concentration at 60 months. Plasma zinc concentration at 24 months was associated with plasma ferritin at 60 months.
Plasma retinol concentration at 60 months	Children belonged to higher trajectory of plasma retinol concentrations during 7 to 24 months had 3.99 µg/dL higher mean plasma retinol at 60 months than the children in lower trajectory.

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
