# Peer review of "Why Do Children in Slums Suffer from Anemia, Iron, Zinc, and Vitamin A Deficiency? Results from a Birth Cohort Study in Dhaka"

_nutrients, 2019, doi:10.3390/nu11123025_

Round 1
Reviewer 1 Report
The article was interesting and the relationship between micronutrients and pathologies of the children sounds great.
We suggestto the Authors to:
1) Explain why they collected data only on retinol,Zinc, Hb and ferritin;
2) Explain why plasma collection for micronutrients was done at 7, 15, 24 and 60 months of age;
3) Try to be "more clear" about the trajectories of plasma micronutrients during the time and with their relationship with "risk of pathologies";
4) Try to insert in the results" a summary table with only the best evidence results about the relationship between micronutrients deficiency and "risk of pathogies";
5) Suggest to the reader "a cut off" trajectories for single nutrients that could be a "predictor" of children pathologies
Author Response
Reviewer 1
The article was interesting and the relationship between micronutrients and pathologies of the children sounds great.
We suggest to the Authors to:
Explain why they collected data only on retinol, Zinc, Hb and ferritin;
Response: The parent study (MAL-ED birth cohort) of the current analysis was originally designed to evaluate the interrelationship between repeated enteric infections and dietary intake as they influence the chances of malnutrition, poor vaccine efficacy, and poor cognitive and behavioral development. In order to do so, a large amount of data on growth, morbidity, pathogen infection, biomarkers of enteropathy, dietary intakes and cognitive function were collected in this large longitudinal study. The protocol was finalized through the formation of cross-site Technical Subcommittees, one of which focused on nutrition measures, and the final protocol was approved by study leadership and the Steering Committee. Thus, the timing of sample collection and the specific assays to be performed were determined for all sites, including Bangladesh. The timing was decided considering other test procedures and subject burden; for example, test burden at 6 months of age was high, and so the blood draw was moved to 7 months. We chose well accepted indicators of micronutrient status for which there is evidence for associations with the outcomes (cognitive development, growth, and vaccine response). Considering the small quantity of blood to be collected from otherwise healthy infants, and prioritization of the sample for vaccine efficacy, a prioritization list was created for the indicators chosen: plasma ferritin, AGP, plasma zinc, and plasma retinol. Hemoglobin was assessed with a finger prick. Urinary iodine excretion was also assessed but is not reported here. The Bangladesh site was able to assay both CRP in addition to AGP at each time point. The original study was designed to follow children from birth to 2 years of age, and multiple micronutrient status measures were desired particularly to conform with cognitive assessments and taking into account testing of vaccine efficacy. After extension of the study, most of the study sites picked up and followed the children until 5 years of age, but only Bangladesh site had funding to complete the assays for plasma micronutrient status at 60 months. Details of the MAL-ED study have been published elsewhere (Caulfield L, CID 2014;59(S4):S248–54).
Explain why plasma collection for micronutrients was done at 7, 15, 24 and 60 months of age;
Response: As described above in response to the 1st comment, the time points for biological sample collections were decided upon by the Vaccine and the Nutrition Technical Subcommittees of the MAL-ED study. These were selected considering time points for vaccine efficacy, burden to families related to other biological sample collections, and cognitive assessments.
3) Try to be "more clear" about the trajectories of plasma micronutrients during the time and with their relationship with "risk of pathologies";
Response: Thank you very much for your valuable suggestion. Trajectories were identified using latent class growth modelling using data from 7 months to 24 months. We tried to explore whether trajectories or patterns of plasma micronutrient concentrations observed in the cohort during 7 to 24 months can predict plasma micronutrient concentrations at 5 years of age. To clarify our findings, we have now provided more information in the results section. Please see page 9- page 12.
4) Try to insert in the results" a summary table with only the best evidence results about the relationship between micronutrients deficiency and "risk of pathologies";
Response: We have included one summary table (Table 6) in the manuscript (page 14).
5) Suggest to the reader "a cut off" trajectories for single nutrients that could be a "predictor" of children pathologies
Response: We have discussed this suggestion and think that to answer this fully would require additional analyses that are beyond the scope of the paper. The paper already contains multiple analyses and important findings. However, to be responsive to this suggestion, we have provided numeric results relating the trajectories of ferritin and retinol to positive outcomes at 60 months. We did not do this for the hemoglobin trajectories because hemoglobin trajectories during 7 to 24 months were unable to predict hemoglobin status at 60 months. We also opted against providing any cut off for zinc as no significant distinct classes existed in this cohort for the progression of concentration of plasma zinc and all individuals followed a similar pattern over the months. Please see page 15.
Reviewer 2 Report
This is an informative study to understand the anemia, iron, ferritin, zinc and vitamin A deficiency in the development of children from under developed regions.
Author Response
Comments and Suggestions for Authors
This is an informative study to understand the anemia, iron, ferritin, zinc and vitamin A deficiency in the development of children from underdeveloped regions.
Response: Thank you very much.
Reviewer 3 Report
The study by Mahfuz et al provides an overview of the micronutrient status among a birth cohort in Dhaka, Bangladesh. This study has public health importance and also tries to establish relations between suspected etiological factors and the observed micronutrient deficiency.
My minor comments can be found below:
In the methods section, the inclusion/exclusion criteria for the cohort are missing. Details should be provided The LCGM trajectories show results only until 24 months. What about 60 months? Details regarding the "infection" context are missing. The authors should consider providing an overview on what kind of infections are they taking into consideration. Are they acute or chronic infections?
Author Response
Comments and Suggestions for Authors
The study by Mahfuz et al provides an overview of the micronutrient status among a birth cohort in Dhaka, Bangladesh. This study has public health importance and also tries to establish relations between suspected etiological factors and the observed micronutrient deficiency.
My minor comments can be found below:
In the methods section, the inclusion/exclusion criteria for the cohort are missing. Details should be provided The LCGM trajectories show results only until 24 months. What about 60 months? Details regarding the "infection" context are missing. The authors should consider providing an overview on what kind of infections are they taking into consideration. Are they acute or chronic infections?
Response: Thank you very much. We have included the inclusion and exclusion criteria for the cohort. Please see page 2.
The second objective of this manuscript is to explore whether early life plasma micronutrient concentrations can predict later age micronutrient status. Therefore, we have calculated the early life trajectories of plasma micronutrients using data from 7 months, 15 months and 24 months plasma concentrations. In this specific analysis, plasma concentrations at 60 months were our outcome variables. Hence, we did not consider 60 months’ data while identifying the trajectories.
Concentrations of plasma micronutrients, particularly ferritin, zinc and retinol are vulnerable to acute phase reactions. We have data of two important acute phase proteins, AGP and CRP. Using data of AGP and CRP by standard methods, the infection variable was created. This represents different stages of acute infection, namely incubation, early convalescence, late convalescence, or no infection. Therefore, for this analysis “infections’ means acute infections. Details can be found on page 3.